# Gap Junction-Mediated Intercellular Communication of cAMP Prevents CDDP-Induced Ototoxicity via cAMP/PKA/CREB Pathway

**DOI:** 10.3390/ijms22126327

**Published:** 2021-06-13

**Authors:** Yeon Ju Kim, Jin-Sol Lee, Hantai Kim, Jeong Hun Jang, Yun-Hoon Choung

**Affiliations:** 1Department of Otolaryngology, Ajou University School of Medicine, Suwon 16499, Korea; yeonju0130@naver.com (Y.J.K.); noto.hantai@gmail.com (H.K.); jhj@ajou.ac.kr (J.H.J.); 2Department of Medical Sciences, Ajou University Graduate School of Medicine, Suwon 16499, Korea; ljs12132@naver.com

**Keywords:** cisplatin, ototoxicity, gap junction, cAMP, forskolin, all-trans retinoic acid

## Abstract

In the cochlea, non-sensory supporting cells are directly connected to adjacent supporting cells via gap junctions that allow the exchange of small molecules. We have previously shown that the pharmacological regulation of gap junctions alleviates cisplatin (CDDP)-induced ototoxicity in animal models. In this study, we aimed to identify specific small molecules that pass through gap junctions in the process of CDDP-induced auditory cell death and suggest new mechanisms to prevent hearing loss. We found that the cyclic adenosine monophosphate (cAMP) inducer forskolin (FSK) significantly attenuated CDDP-induced auditory cell death in vitro and ex vivo. The activation of cAMP/PKA/CREB signaling was observed in organ of Corti primary cells treated with FSK, especially in supporting cells. Co-treatment with gap junction enhancers such as all-trans retinoic acid (ATRA) and quinoline showed potentiating effects with FSK on cell survival via activation of cAMP/PKA/CREB. In vivo, the combination of FSK and ATRA was more effective for preventing ototoxicity compared to either single treatment. Our study provides the new insight that gap junction-mediated intercellular communication of cAMP may prevent CDDP-induced ototoxicity.

## 1. Introduction

The sensory epithelium of the inner ear contains mechanosensory cells, known as hair cells (HCs), each of which is surrounded by non-sensory supporting cells (SCs). SCs are directly connected to adjacent SCs via gap junctions. Gap junctions, consisting of connexin (Cx) proteins, are specialized intercellular channels between adjacent cells that allow the passage of small molecules, including second messengers, ions, and metabolites of up to 1 kDa. In the inner ear, the elaborate network of gap junctions is essential to hearing function due to its cycling of potassium ions to maintain high endocochlear potential and fluid homeostasis [1,2]. The essential role of gap junctions in hearing has been shown in numerous genetic analyses and functional linkage studies between the mutation of Cx genes and hearing dysfunction in human and animals [3,4]. Fibrocytes in the lateral wall (LW) of the cochlea are also involved in potassium ion exchange via gap junctions during auditory transduction. Five different types of fibrocytes are present in the LW, distinguished by their location, structural characteristics, and levels of enzymes mediating ion transport [5]. Despite the significance of gap junctions to hearing function, little is known about the biological molecules that are communicated through cochlear gap junction networks and affect hearing loss caused by external sources of damage, including ototoxic drugs and noise.

The exchange of small molecules via gap junctions is also thought to be involved in the response to external stimuli such as cytotoxic chemicals and radiation. The modulation of gap junctional intercellular communication (GJIC) or Cx genes has been shown to regulate the transmission of signaling molecules in various pathological conditions and diseases, which in turn could cause cell death or survival, depending on the tissue type and extent of injury. Several studies have suggested that Ca^2+^, inositol 1,4,5-triphosphate, cyclic adenosine monophosphate (cAMP), guanosine monophosphate, and diacylglycerol (DAG) may be some of the small molecules that pass through gap junctions and are responsible for cytotoxic effects [6,7,8].

Cisplatin (CDDP) is one of the most widely used chemotherapeutic agents for the treatment of many types of solid cancers. However, its use causes irreversible hearing loss in 40–80% of patients [9]. Despite the high incidence of permanent hearing loss, there are no drugs approved by the Food and Drug Administration (FDA) for the prevention of CDDP-induced ototoxicity. Systemic injection of CDDP damages HCs and SCs within the organ of Corti [10]. CDDP is transferred into cells through transporters, including copper transporter 1 and organoid cation transporter 2, and causes cytotoxic effects through DNA cross-linking and the production of reactive oxygen species (ROS), leading to oxidative damage and cell death [11]. In spite of numerous studies on the mechanism of CDDP-induced cytotoxicity, the ototoxic pathophysiology remains unclear and most studies have focused on intracellular mechanisms.

Our previous studies showed that the pharmacological regulation of gap junctions prevents CDDP-induced ototoxicity [12]. However, the question of which small molecules pass through gap junctions and regulate cell death or survival remains open.

In this study, we aimed to identify specific molecules that pass through gap junctions during the process of CDDP-induced cell death and suggest new mechanisms for the prevention of hearing loss. We then assessed the hearing function of CDDP-induced ototoxic animal models treated with candidate compounds.

## 2. Results

### 2.1. Increasing cAMP Protects against CDDP-Induced Auditory Cell Death via Activation of the cAMP/PKA/CREB Pathway

To identify the which of the small molecules that regulate cell death or survival through gap junctions are also involved in CDDP-induced cell death, we first tested cell viability using the following compounds: the IP_3_ inhibitor 2-APB, the mitochondrial Ca^2+^ uptake inhibitor Ru360, the cAMP inhibitor SQ22536, the cAMP inducer forskolin (FSK), the cGMP-dependent protein kinase inhibitor KT5823, and the DAG inhibitor DAGi. The drug doses were selected based on a pilot study and the literature [13,14,15]. The HEI-OC1 cell line was pre- and co-treated with these drugs with or without CDDP for 24 h. We found that FSK, an inducer of cAMP, significantly increased cell viability (Figure 1A). When treated with various concentrations of FSK in combination with CDDP, HEI-OC1 cell viability increased in a dose-dependent manner. SQ22536, an inhibitor of cAMP, markedly reduced the FSK-induced elevation of cell viability (Figure 1B). In addition, rolipram, a selective inhibitor of PDE4, and the cAMP analogue 8-Br-cAMP suppressed cell death (Figure 1C) and the cleavage of both poly-adenosine diphosphate ribose polymerase (PARP) and caspase-3 (data not shown). The intracellular cAMP concentration was significantly elevated in the FSK treatment (Figure 1D). These results indicate that cAMP signaling protects against CDDP-induced auditory cell death. Next, we investigated the functional impact of cAMP induction by FSK using primary cells of the organ of Corti (OC). We found that FSK treatment reduced the loss of HCs with CDDP treatment in the organotypic culture of OC (data not shown). cAMP/protein kinase A (PKA)/cAMP-response element binding protein (CREB) signaling initiates a downstream pathway that regulates various cellular responses, including survival, proliferation, metabolism, cell cycle, and ion channels [16]. Upon the binding of cAMP to the PKA regulatory subunits, PKA catalyzes substrate phosphorylation. To detect PKA activity, we used a phospho-PKA (p-PKA) substrate antibody. As shown in Figure 1E, phosphorylation of PKA substrate was induced by FSK with or without CDDP. The FSK-induced increase in PKA–substrate phosphorylation was reduced with the addition of SQ22536 and H89, specific inhibitors of cAMP and PKA, respectively, while caspase-3 activity associated with apoptosis was increased by these additions (Figure 1G).

The phosphorylation of cAMP response element-binding protein (CREB) at Ser 133 is a direct target of PKA downstream of G protein-coupled receptors in the Gαs-cAMP signaling pathway [17], which regulates various genes as a transcription factor. We found an increase in the phosphorylation level of CREB with FSK treatment, accompanied by reduced cleavage of caspase-3 and PARP (Figure 1F and Appendix A). Positive immunoreactivity to phospho-CREB (p-CREB) was localized in the SCs, including in Dieter’s cells and Hensen’s cells (Figure 1H). These results indicate that cAMP signaling prevents CDDP-induced auditory cell death via a PKA/CREB-mediated pathway.

### 2.2. The Gap Junction Enhancer Retinoic Acid Improves the Protective Effect of FSK against CDDP

To test whether gap junction channels could influence cAMP-dependent cell survival, we performed combined treatment with FSK, a gap junction enhancer (PQ1, ATRA), and a gap junction inhibitor (18α-GA, oleamide) under CDDP exposure. Co-treatment with both PQ1 and ATRA resulted in significantly higher cell viability than non-treatment. Statistical significance for the effect of ATRA was stronger than that of PQ1 (data not shown). We next investigated whether the influence of cAMP in response to CDDP could be further increased by the gap junction enhancer ATRA. Primary OC cells were treated with ATRA, FSK, and CDDP individually or in combination. The combinational treatment of 50 μM ATRA and FSK both increased p-CREB and decreased cleaved PARP and caspase-3 compared to only FSK with CDDP (Figure 2A).

Using the FRAP assay, we found 13.1% recovery of fluorescent signals in DMSO-treated HEI-OC1 cells within 500 s after photobleaching, and the recovery rate increased to 19.4% in cells treated with ATRA. In contrast, non-contact sequestered HEI-OC1 cells exhibited reduced fluorescence, and no changes were observed in ATRA-treated non-contact cells (Figure 2B,C). ATRA has been reported to function as a gap junction potentiator by upregulating Cx proteins [18]. We determined that the expression of Cx43, Cx30 and Cx26 was significantly increased by ATRA in a dose-dependent manner (range 25–150 µM) (Appendix A). Immunostaining revealed the increased size of Cx43 gap junction plaques (Figure 2D, Appendix A). Previous studies have shown that CDDP treatment inhibits the function of gap junctions and Cx43 expression [19,20]. Compared to the DMSO control, the expression of Cx43 was reduced with CDDP treatment, whereas ATRA treatment significantly increased Cx43 expression, even when combined with CDDP exposure (Figure 2E). Our results indicate that cAMP-dependent signaling can be potentiated by gap junctions to prevent CDDP-induced cell death.

### 2.3. Combinational Treatment of FSK and ATRA Prevents CDDP-Induced Hearing Loss

We investigated the effect of combinational treatment with FSK and ATRA against CDDP-induced hearing loss. For in vivo experiments, SD rats (n = 89) were given a single intraperitoneal (IP) injection of CDDP (16 mg/kg). FSK (25 µM) and ATRA (30 µM) were administered with an intratympanic (IT) injection into the right ears four times over 5 days, including before and after CDDP IP injection (Figure 3A). All left ears were injected with normal saline (NS) as a CDDP control. The relative threshold shift, which is the level of each rat’s right ear minus that of the left ear, was preserved slightly in the FSK treatment (2.3 ± 2.2 dB at 8 kHz, 5.0 ± 3.8 dB at 16 kHz, and 4.1 ± 3.4 dB at 32 kHz), ATRA (3.6 ± 2.0 dB at 8 kHz, 4.5 ± 1.6 dB at 16 kHz, and 4.5 ± 1.6 dB at 32 kHz), and FSK + ATRA (8.2 ± 5.1 dB at 8 kHz, 5.9 ± 1.9 dB at 16 kHz, and 5.0 ± 2.3 dB at 32 kHz), compared to the left ear treated with NS. However, no statistically significant difference was observed due to differing susceptibility to CDDP (Figure 3B).

We compared the hearing loss prevention rate, and found mild, moderate or strong prevention of hearing loss in 0 of 15 rats (0%) at 8, 16, and 32 kHz in the CDDP group; 4 of 13 (31%) at 8 kHz, 5 of 13 (62%) at 16 kHz, and 4 of 13 (69%) at 32 kHz in the FSK group; 3 of 11 (27%) at 8 kHz, 6 of 11 (55%) at 16 kHz, and 5 of 11 (45%) at 32 kHz in the ATRA group; and 5 of 11 (45%) at 8 kHz, 7 of 11 (64%) at 16 kHz, and 5 of 11 (45%) in the FSK + ATRA combination group (Figure 3C). The FSK + ATRA combination group showed a significantly higher prevention rate than the FSK or ATRA groups. In particular, the strong prevention rate was significantly higher in the FSK + ATRA combined group at 8 kHz (Figure 3C). Overall, the prevention rate was significantly higher when combining FSK and ATRA at 8 and 16 kHz (Figure 3D).

### 2.4. Treatment with FSK and ATRA Prevents CDDP-Induced Loss of HCs, SGNs and LW Fibrocytes by Improving cAMP and Gap Junction Function

We examined the histological features of cochlear sections by H&E staining and immunohistochemistry. First, cell counts and nuclear morphology were evaluated in H&E-stained images. In the OCs, CDDP treatment significantly decreased HCs and SCs (Appendix A). However, HCs did not differ significantly among the CDDP-treated experimental groups, whereas SCs were increased significantly in the FSK + ATRA group compared to the CDDP group (Appendix A). In transverse sections of the cochlea, few HCs were visible. Therefore, whole-mount cochleae were stained with the hair cell-specific markers myosin VIIa and phalloidin to compare hair cell preservation between the CDDP and FSK + ATRA groups. Rats treated with FSK + ATRA had significantly more HCs than the NS group (Appendix A). Quantitative analysis confirmed that the loss of SGNs was significantly inhibited in the FSK and FSK + ATRA groups compared to the NS group. Many more SGNs were present in the FSK + ATRA group than in the FSK group (Appendix A). CDDP induced apoptosis, characterized by cell shrinkage, nuclear condensation, and fragmentation [21]. Nuclear condensation was observed in the CDDP group but was noticeably lower in the FSK, ATRA, and FSK + ATRA groups (Appendix A). Fibrocytes that play a role in ion transport and the generation of endocochlear potential can also be damaged by CDDP. We found no significant loss of fibrocytes after CDDP treatment, but there was a significant increase in type I fibrocytes in the FSK + ATRA group (Appendix A).

We next examined the expression of p-CREB, p-PKA, Cx26, and Na^+^/K^+^-ATPase in cochlear tissues to investigate whether the preventive effects of FSK and ATRA on CDDP-induced ototoxicity were mediated by cAMP and gap junction functions. In the cochlea, p-CREB was observed in SCs of the OC, SGNs, and LW fibrocytes. In the OC and LW, the expression of p-CREB was lower in the CDDP group than in the FSK and FSK + ATRA groups (Figure 4A,C). p-PKA immunostaining was observed throughout the HCs and SCs of the OC, SGNs, and LW fibrocytes. The p-PKA level decreased after CDDP treatment, but increased significantly after the drug treatments compared to the controls (Figure 4D–F). Furthermore, p-PKA expression was much higher after treatment with the ATRA + FSK combination than with FSK alone. To evaluate the effect of ATRA on cochlear Cx expression, we compared the expression of Cx26 protein among the groups. In the cochlea, Cx26 was highly expressed in SCs of the OC, SGNs, and LW fibrocytes (Figure 5). The CDDP group expressed a significantly lower level of Cx26 compared with the controls. Cx26 expression was significantly increased in the FSK, ATRA and FSK + ATRA groups compared to the NS group after CDDP treatment (Figure 5A–C,E). Na^+^/K^+^-ATPase is expressed in the marginal cells of the stria vascularis and fibrocytes in the LW, and participates in the active transport of Na^+^ and K^+^ to maintain cochlear function [22]. CDDP reduces the levels of Na^+^/K^+^-ATPase, which might disrupt endocochlear potential [23]. We also found that CDDP reduced the Na^+^/K^+^-ATPase level (Figure 5D,E). IT injection of ATRA or FSK + ATRA significantly increased Na^+^/K^+^-ATPase immunoreactivity compared to the NS-treated animals. Together, our results suggest that the activation of the cAMP signaling pathway and gap junction function are involved in cochlear cell survival in CDDP-induced ototoxicity.

### 2.5. FSK and ATRA Do Not Attenuate the Anti-Tumor Properties of CDDP

Establishing whether the administration of FSK and ATRA had a deleterious effect on the anti-tumor efficacy of CDDP was critically important. ATRA and its analogs, retinoids, along with retinoic acid receptors are well known to have tumor-suppressive effects [24,25], although the anti-tumor effects of FSK and the FSK + ATRA combination are unclear. Therefore, we tested the anti-tumor effect of FSK in vivo and the combination effect in vitro in this study. To determine whether FSK affects the anti-tumor efficacy of CDDP, a tumor xenograft model was employed. A549 human lung adenocarcinoma cells were subcutaneously transplanted into male nude mice. After tumor establishment, the mice were randomized into the following four groups according to IP injection: vehicle (NS), FSK, CDDP, and CDDP + FSK. CDDP significantly decreased the tumor growth (Figure 6A), and co-treatment with FSK did not attenuate the anti-tumor effect of CDDP. Consistently, the CDDP + FSK combined treatment showed no change in the proliferation marker Ki-67 or the apoptosis marker cleaved caspase-3 (Figure 6G) compared to the CDDP group. Next, an in vitro study was performed to compare the effect of combined treatment with FSK and ATRA against CDDP-induced damage. We used the auditory cell line HEI-OC1, the cancer cell line A549, and HeLa cells. As expected, treatment with FSK and ATRA decreased levels of cleaved PARP and phospho-γ-H2AX, which are indicators of apoptosis and DNA damage, respectively (Appendix A), and the combined treatment showed a potentiating effect. In addition, the downregulation of p21 and the induction of cyclin D1, which are required for cell survival, were observed in the treatment group. Conversely, the expression levels of cleaved PARP and phospho-γ-H2AX were enhanced by combined treatment in A549 cells, but were unchanged in HeLa cells (Appendix A). Moreover, no changes in p-PKA substrate and Cx43 expression were observed among groups. Therefore, combination treatment with FSK and ATRA does not interfere with the anti-tumor properties of CDDP, and potentiates the protective effect of FSK against CDDP-induced ototoxicity.

## 3. Discussion

In this study, we show that FSK-inducible cAMP activation prevents CDDP-induced auditory cell death. The protective effect of FSK against ototoxicity is mediated by the activation of the cAMP/PKA/CREB pathway in SCs of the organ of Corti. The upregulation of gap junctions by ATRA potentiated cAMP-dependent signaling by propagating signals through interconnected cells. Combined treatment with FSK and ATRA exhibited a preventive effect against hearing loss in CDDP-induced ototoxic animal models without impacting the anti-tumor effect of CDDP in cancer cells.

cAMP is a second messenger that regulates diverse cellular processes; in particular, cAMP-induced PKA/CREB activation governs apoptosis prevention through the direct regulation of transcriptional events that are essential to cell survival, such as those leading to the expression of Bcl-2, cyclin D1, and cyclin A [26]. Our data obtained using in vitro and in vivo models demonstrate for the first time that the cAMP/PKA/CREB pathway can inhibit CDDP-induced ototoxicity. A similar study has shown that rolipram, which acts as a specific inhibitor of type IV phosphodiesterase (PDE4) leading to increased intracellular levels of cAMP, improves the survival of SGNs via the cAMP/PKA and MAP-kinase pathway when SGNs are damaged following cochlear implant [27].

FSK, a natural compound, is a diterpene produced by the roots of the Indian plant *Coleus forskohlii*. FSK has a long history of medical application and its safety has been confirmed by many researchers [28]. In practice, FSK has been used for the treatment of asthma, glaucoma, hypertension, cancer, diabetes, and obesity [29]. Recent research has shown that FSK may play a protective role in CDDP-induced ototoxicity by inhibiting the mitochondrial apoptotic pathway and ROS production [30]. Under noise exposure conditions, treatment with FSK attenuated noise-induced losses of outer hair cells by blocking the ROS/AMPKα-dependent pathway [31]. We initially identified cAMP and its activator in an attempt to find small molecules that not only are capable of crossing gap junction channels, but that also regulate auditory cell death related to CDDP. We also showed that cAMP-dependent signaling is enhanced by ATRA, a gap junction activator.

Increasing cell-to-cell transfer of cAMP through gap junctions can enhance responses to internal and external stimuli [32,33,34]. Using the gap junction blockers 18α-GA and Cx43 siRNA, a study showed that cAMP levels in vascular endothelial cells (VECs) and the transfer of cAMP from VECs to vascular smooth muscle cells (VSCMs) were mediated by myoendothelial gap junctions after hypoxia associated with angiopoietin-2 treatment [34]. Moreover, the major osteoblast gap junction protein Cx43 can transfer cAMP between osteoblasts in a cell-contact-dependent manner, and the transfer of cAMP through Cx43 gap junctions synergistically affects *RANKL* and *sclerostin* gene expression, which are key regulators of bone homeostasis [32]. In the cochlea, gap junctions are essential to normal hearing. Mutations in *Gja1* (Cx43), *Gjb2* (Cx26), *Gjb3* (Cx31), and *Gjb6* (Cx30), which constitute gap junctions, can cause non-syndromic hearing loss [4]. Interestingly, the overexpression of Cx26 rescues hearing in the human *Gjb6* null mutation mouse model [35]. Furthermore, the upregulation of Cx26 expression by ATRA significantly improved hearing in Cx30 knockout mice by 10 dB. These results suggest that pharmacological agents that enhance Cx expression could restore hearing and that 30 μM ATRA injection through the round window membrane sufficiently increases Cx proteins in the cochlea. In the present study, we show the otoprotective effect of IT injection of 30 μM ATRA alone in CDDP-induced ototoxic animal models.

In this study, we showed that systemic injection of CDDP increases the phosphorylation of CREB, which was further increased by FSK treatment. Using OC explant cultures and cochlear tissues, we observed increased phosphorylation of CREB in SCs. These results are similar to reports of other experiments where FSK-induced cAMP accumulation occurred predominantly in SCs. FSK-induced SC regeneration is mediated by increases in the expression of cAMP and the MAP-kinase/ERK member B-Raf in avian SCs [36]. Similarly, pCREB is involved in compensatory proliferation related to cell regeneration and survival. As Cx proteins were restricted to SCs, Cx gap junctions may synergistically enhance the cAMP-dependent signals through the propagation of direct cell-to-cell communication. We propose that FSK-induced CREB phosphorylation can promote the survival of cochlear SCs, and that this effect can be enhanced by increasing Cx levels. Unlike aminoglycoside antibiotic-induced ototoxicity, which only damages HCs in the OC, CDDP damages both HCs and SCs. The survival of SCs could indirectly contribute to HC survival by maintaining the cytoarchitecture of the sensory epithelium [37], and by secreting cytoprotective molecules such as HSP70, HSP60, ERK, and Wnt/β-catenin to nearby HCs [37,38,39,40]. Interestingly, a recent study revealed that HSP70-dependent paracrine intercellular communication in SCs protects HCs and is mediated by exosomes [41]. Furthermore, exosomes may functionally cooperate with the cAMP or PKA pathway [42,43]. Further research is needed to demonstrate how the presence of SCs affects HC survival.

Previous studies have shown that FSK exerted anti-tumor effects with no toxicity to normal cells and increased the anti-tumor activities of chemotherapeutic drugs including dexamethasone, gemcitabine, and everolimus when used in combination [44,45]. Our results showed that FSK did not affect the anti-tumor effects of CDDP on lung cancer in a mouse xenograft model. However, the phosphorylation of PKA by FSK was observed in both the A549 and HeLa cell cancer cell lines. The mechanism through which FSK protects auditory cells without affecting cancer cells remains unclear. ATRA, which we used as a gap junction enhancer, is known to inhibit the growth of tumor cells, leading to differentiation, apoptosis and anti-tumor gene expression (e.g., *p53*, *Foxo1*, *Foxo3*, and *p21*) [46,47]. Moreover, several studies have shown that combination treatment with retinoic acid enhances the effect of CDDP on cancer cells by facilitating apoptosis, cell cycle arrest [48], and the downregulation of thiol antioxidant defense [49], as well as by inducing the differentiation of tumor-initiating cells [25]. ATRA also prevented CDDP-induced ototoxicity without blocking chemotherapeutic efficacy in an in vitro model. Studies have revealed that ATRA negatively regulates oxidative stress related to CDDP damage that causes nephrotoxicity and testicular toxicity [50,51]. Additionally, treatment with ATRA had protective effects against ethanol-induced embryonic HC loss [52]. We could not rule out the possibility that the protective effect of ATRA against CDDP-induced ototoxicity may be mediated by another mechanism.

In conclusion, we suggest that cAMP is a biologically important second messenger communicated by SCs through gap junctions. Combined treatment with FSK and ATRA may provide a promising candidate for preventing CDDP-induced ototoxicity during CDDP-based cancer therapy.

## 4. Materials and Methods

### 4.1. Organ of Corti Explant Culture and Treatment

All animal experiments performed for this study were approved by the Institutional Animal Care and Use Committee (IACUC) of Ajou University (IACUC-2019-0028). Sprague Dawley (SD) rats (Daehan Bio-link, Cheongju, Korea) were decapitated on postnatal day 5. The cochleae were removed, the LWs were dissected away, and the organs of Corti (OCs) were placed in a tissue culture plate. OC explants were maintained in Dulbecco’s modified Eagle medium (DMEM; Gibco-BRL, Grand Island, NY, USA) containing 10% fetal bovine serum (FBS; Gibco-BRL, NY, USA) and 0.06 mg/mL penicillin (Sigma-Aldrich, Steinheim, Germany) at 37 °C under 5% CO_2_. After 24 h, OC explants were exposed to medium containing CDDP (15 μM, Sigma-Aldrich, St. Louis, MO, USA), forskolin (FSK; 25 μM, Sigma-Aldrich, St. Louis, MO, USA), and all-trans retinoic acid (ATRA, 50 μM, Sigma-Aldrich, St. Louis, MO, USA) for 24 h.

### 4.2. House Ear Institute-Organ of Corti 1 (HEI-OC1) and OC Primary Cell Culture and Treatment

HEI-OC1 cells, a conditionally immortalized OC cell line, were grown in DMEM (Gibco-BRL, Grand Island, NY, USA) containing 10% FBS (Gibco-BRL, NY, USA) at 33 °C under 10% CO_2_. For OC primary cell culture, the cochlea was dissected from SD rats on postnatal day 5. The LW and spiral ganglion neurons (SGNs) were removed, and OCs were enzymatically digested with 0.25% trypsin–EDTA (Gendepot, Barker, TX, USA) in phosphate-buffered saline (PBS) for 5 min at 37 °C. After centrifugation at 3000 rpm for 3 min to remove the supernatant, OC primary cells were plated on cell culture plates and incubated in DMEM (Gibco-BRL, Grand Island, NY, USA) containing 10% FBS (Gibco-BRL, NY, USA) and 0.06 mg/mL penicillin (Sigma-Aldrich, Steinheim, Germany) at 37 °C under 5% CO_2_.

Cells were pre-treated with 2-aminoethoxydiphenyl borate (2-APB; Sigma-Aldrich, St. Louis, MO, USA), Ru360 (EMD Biosciences, San Diego, CA, USA), FSK (Sigma-Aldrich, St. Louis, MO, USA), SQ22536 (Sigma-Aldrich), KT5823 (Sigma-Aldrich, St. Louis, MO, USA), diacylglycerol kinase inhibitor (DAGi; Sigma-Aldrich, St. Louis, MO, USA), ATRA (Sigma-Aldrich, St. Louis, MO, USA), H89 (Selleckchem, Shanghai, China), 18 alpha-glycyrrhetinic acid (18α-GA, Sigma-Aldrich, St. Louis, MO, USA), and oleamide (Sigma-Aldrich, St. Louis, MO, USA) for 2 h before CDDP (15 μM, Sigma-Aldrich, St. Louis, MO, USA) application.

### 4.3. Enzyme-Linked Immunosorbent (ELISA) Assay

ELISA was performed using a commercial cAMP ELISA assay kit (R&D systems, Minneapolis, MN, USA), according to the manufacturer’s instructions. In brief, primary antibody solution was added to wells and incubated for 1 h at room temperature on a horizontal orbital microplate shaker. Then, each well was aspirated and washed three times with washing buffer, and cAMP conjugate was added, followed by addition of 100 µL of standard, control, or sample to each well. The wells were covered with an adhesive strip and incubated for 2 h at room temperature on the microplate shaker. Following washing, 200 µL of substrate solution was applied to each well and the color was allowed to develop for 30 min. Using a microplate reader (iMark Microplate Absorbance Reader; Bio-Rad Laboratories, Inc., Hercules, CA, USA), absorbance readings at 450 nm were taken.

### 4.4. Fluorescence Recovery after Photobleaching (FRAP)

The FRAP assay was carried out using a monolayer of living cells, as described previously [19]. Briefly, cells were plated in a 35-mm glass bottom confocal dish (*SPL* Life Science, Gyeonggi-do, Korea). After washing with PBS, calcein acetoxymethyl derivative (calcein AM; 1 µM; Molecular Probes, Eugene, OR) in 4-(2-hydroxyethyl)-1-piperazineethanesulfonic acid (HEPES; GIBCO, Carlsbad, CA) was added and incubated for 15 min. A cell in direct contact with 4–5 neighboring cells within a cluster was selected as the region of interest (ROI) for junctional analysis. The ROI was photobleached with a short burst of intense light from an argon laser, and its fluorescence was monitored every 10 s for 500 s. The same laser intensity and monitoring time were used for FRAP analysis of all samples. All data were normalized to 100 based on pre-bleaching fluorescence intensity.

### 4.5. Western Blot Analysis

Protein samples were separated using sodium dodecyl sulfate polyacrylamide gel electrophoresis (SDS-PAGE). After electrophoresis, the proteins were transferred onto a polyvinylidene fluoride membrane (Millipore, Billerica, MA, USA), and blocked with 5% skim milk in PBS containing 0.05% Tween 20 (PBST) for 1 h. The membranes were incubated with primary antibodies at 4 °C overnight. After washing with PBST, the membranes were probed with horseradish peroxidase-conjugated secondary antibodies (GenDepot, Katy, TX, USA) for 1 h at room temperature. Following washing of the membrane, the immunoblot bands were visualized using ECL Western Blotting Substrate (Thermo Fisher Scientific, Inc, Grand Island, NY, USA). The following antibodies were used: anti-phospho-PKA substrate (#9624, Cell Signaling, Danvers, MA, USA), anti-cyclin D1 (#55506, Cell Signaling), anti-phospho-CREB (#9198, Cell Signaling), anti-total-CREB (#9197, Cell Signaling), anti-connexin 43 (#3512, Cell Signaling), anti-cleaved caspase-3 (#9661, Cell Signaling), and anti-cleaved PAPR (ab32064, Abcam, Cambridge, UK).

### 4.6. Hematoxylin and Eosin (H&E) Staining and Histological Analysis

Cochleae were harvested and fixed with 4% paraformaldehyde at 4 °C for 2 days. They were washed with PBS and decalcified using Calci-Clear Rapid (National Diagnostics, Atlanta, GA, USA) for 5 days. Decalcified cochleae tissues were embedded in paraffin using an automated tissue processing system. The paraffin blocks were sliced into 5-μm-thick sections. The slides were dried for 30 min on a slide warmer at 60 °C, deparaffinized, rehydrated in a graded alcohol series, and then stained with H&E. The percentages of HCs, SGNs, and fibrocytes in the middle turn of the cochleae of at least five ears were scored, and the average was calculated.

### 4.7. Immunohistochemistry

After deparaffinization and rehydration, antigen retrieval was performed using IHC-Tek ^™^ Epitope Retrieval Solution (IHC WORLD, Ellicott City, MD, USA) and by boiling (95 °C) the slides in plastic Coplin jars for 20 min. After cooling at room temperature for 30 min, endogenous peroxidase was blocked in 3% H_2_O_2_ in methanol at room temperature for 10 min. The slides were incubated with 1% bovine serum albumin in PBST for 1 h. Anti-phospho-CREB (#9198, Cell Signaling), anti-total-CREB (#9197, Cell Signaling), anti-phospho-PKA (#9624, Cell Signaling), anti-Cx26 (51-2800, Zymed Laboratories), and anti-sodium potassium ATPase (ab7671, Abcam) antibodies were applied overnight at 4 °C. The slides were washed and incubated with peroxidase-conjugated secondary antibodies for 1 h at room temperature. The sections were washed three times with PBST buffer and reacted with 3,3-diaminobenzidine (DAB) using a DAB substrate kit (ab64238). Sections were counterstained with hematoxylin. Images were obtained by bright field microscopy (Olympus, Tokyo, Japan). Chromogen intensity was quantified as described [53]. The mean gray values in ROIs were averaged over five independent sections per group (n = 5) and presented as the relative chromogenic intensity compared with the controls.

### 4.8. Immunofluorescence

OC explants or cells were fixed with 4% paraformaldehyde (Biosesang, Gyeonggi-do, Korea) for 15 min at room temperature and permeabilized using 0.2% Triton-X 100 in PBS (PBST). After blocking with 1% bovine serum albumin in 1× PBST, samples were incubated with primary antibodies at 4 °C for 24 h. The primary antibodies were anti-phospho-CREB (#9198, Cell Signaling), anti-total-CREB (#9197, Cell Signaling), anti-connexin 43 (#3512, Cell Signaling), and anti-SOX2 (1:500, Cell Signaling; #4900). Samples were washed thoroughly and incubated with secondary antibodies tagged with fluorescein isothiocyanate (FITC) or cyanine 3 (Cy3; Jackson ImmunoResearch Laboratories, West Grove, PA, USA) for 1 h at room temperature. F-actin was stained using phalloidin–Texas red (Invitrogen, Molecular Probes, Carlsbad, CA, USA), and nuclei were counterstained with 4′6,-diamidino-2-phenylindole (DAPI; Invitrogen). Coverslips were mounted onto slides with mounting medium (Vector Laboratories, Burlingame, CA, USA). The immunostained cells were observed using a Zeiss LSM 700 confocal microscope (Carl Zeiss Meditec, Jena, Germany).

### 4.9. Animal Experiments

Male SD rats at 7 weeks of age were purchased from Daehan Bio-link (Cheongju, Korea). SD rats were housed at 22 ± 2 °C under a standard 12:12 h light–dark cycle for at least 1 week of acclimation. Animals were randomly divided into five groups (n = 11 per group), which were (a) control, (b) CDDP + normal saline (NS), (c) CDDP + FSK, (d) CDDP + ATRA and (e) CDDP + FSK + ATRA. FSK (25 µM, 20 µL), ATRA (30 µM, 20 µL), and combinations of these drugs were administered three times at an interval of 24 h via intratympanic (IT) injection into the right ear. An equivalent volume of 0.9% NS was injected into the left ear. CDDP was dissolved in NS at a low concentration (0.6 mg/mL) for hydration therapy, and was administered via intraperitoneal injection (IP) at a dose of 16 mg/kg. Pre- and post-hydration was performed three times at an interval of 48 h via IP injection of 25 mL/kg NS. Auditory brainstem response (ABR) thresholds were measured on day 5 after CDDP injection (Figure 3A).

### 4.10. Auditory Brainstem Response (ABR) Test

ABRs were measured with the BioSig 32 System (Tucker-Davis Technologies, Gainesville, FL, USA). Each rat was anesthetized with an injection of Zoletil 50 (Virbac Laboratoires, Carros, France) and Rompun 2% (Bayer Korea, Ansan, Korea), and then placed in a sound-shield booth, which excludes outside noise. ABRs were transcutaneously recorded using sterile electrode needles. The thresholds were determined as the lowest intensity level at which a clear waveform was visible in the trace produced. Frequency-specific ABRs in response to tone burst stimuli were recorded at 8, 16, and 32 kHz. Rats were excluded if their hearing thresholds exceeded 15 dB at frequencies of 16 or 32 kHz prior to the experiments. All ABRs were analyzed in a blind fashion.

### 4.11. Tumor Xenograft Mouse Model

Six-week-old male Balb/c-nu/nu nude mice were used. The A549 cells were purchased from ATCC (Manassas, VA, USA), and incubated in DMEM containing 10% FBS (Gibco-BRL) and 5% penicillin/streptomycin at 37 °C under 5% CO_2_. Nude mice were inoculated with 2 × 10^6^ A549 cells in 50 µL of medium via subcutaneous injection in the right and left dorsal flanks for 5 consecutive days. After inoculation, tumor growth was monitored for 1 week using a Vernier caliper. Animals were randomly divided into four groups for treatment with NS, FSK (2.5 mg/kg), CDDP (8 mg/kg), or CDDP + FSK. The drugs were IP injected for 5 days followed by 1 week of rest, and this sequence was repeated twice. During treatment, body weight and tumor size were measured weekly.

### 4.12. Statistical Analysis

For comparisons between two groups, the non-parametric Mann–Whitney U test was used. Chi-square testing was used to compare the prevention rate (no prevention/mild prevention/moderate prevention/strong prevention) of hearing loss between each treatment (FSK, ATRA, or FSK/ATRA combined) and the NS treatment. All statistical analyses were performed using IBM SPSS Statistics for Windows software (version 23.0; IBM Corp.; Armonk, NY, USA). *p*-value < 0.05 indicates statistical significance.

## Figures and Tables

**Figure 1 ijms-22-06327-f001:**
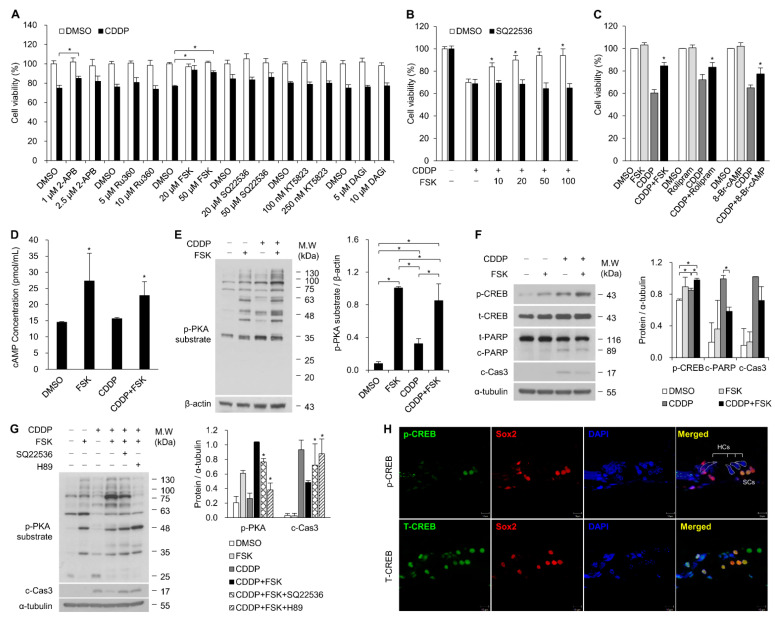
Increased cAMP due to FSK protects against CDDP-induced auditory cell death through upregulation of the cAMP/PKA/CREB pathway. All bar graphs show mean ± standard deviation (* *p* < 0.05). (**A**) Screening for compounds that can pass through gap junctions and inhibit CDDP-induced ototoxic cell death. HEI-OC1 cells were incubated with 2-ABP, Ru360, KT5823, FSK, SQ22536, and DAGi for 2 h and then co-treated with 15 μM CDDP for 24 h. Cell viability was quantified by water-soluble tetrazolium salt (WST-1) assay. (**B**) HEI-OC1 cells were treated with various concentrations of FSK along with CDDP or CDDP + SQ22536. (**C**) Effects of the cAMP-elevating compounds FSK (adenylyl cyclase activator), rolipram (PDE4 inhibitor), and 8-Br-cAMP (cAMP analogue) against CDDP-induced effects on cell viability. HEI-OC1 cells were incubated with 20 μM FSK, 20 μM rolipram, and 0.1 mM 8-Br-cAMP for 2 h and then co-treated with 15 μM CDDP for 24 h. * *p* < 0.05 relative to CDDP treatments. (**D**) Intracellular cAMP levels were measured using ELISA in HEI-OC1 cells treated with 20 μM FSK and 15 μM CDDP. (**E**,**F**) Organ of Corti primary cells were treated with 25 µM CDDP after pretreatment with 25 µM FSK for 24 h. The p-PKA level was analyzed by Western blot and normalized to β-actin. The phosphorylated and total forms of CREB, cleaved PARP, and caspase-3 were detected via Western blot. p-CREB was normalized to total CREB, while cleaved PARP and caspase-3 were normalized to α-tubulin levels. (**G**) Western blot analyses of p-PKA and cleaved caspase-3 using organ of Corti primary cell lysates derived from cultures treated with 25 µM FSK for 24 h with the adenylate cyclase inhibitor SQ22536 (50 µM) or PKA inhibitor H89 (5 µM). Significant differences were found between CDDP + FSK and CDDP + FSK + SQ22536 or H89 treatments. (**H**) Immunohistochemistry of phosphorylated and total CREB in the organ of Corti at 7 weeks. The white dotted lines indicate the contours of the HCs.

**Figure 2 ijms-22-06327-f002:**
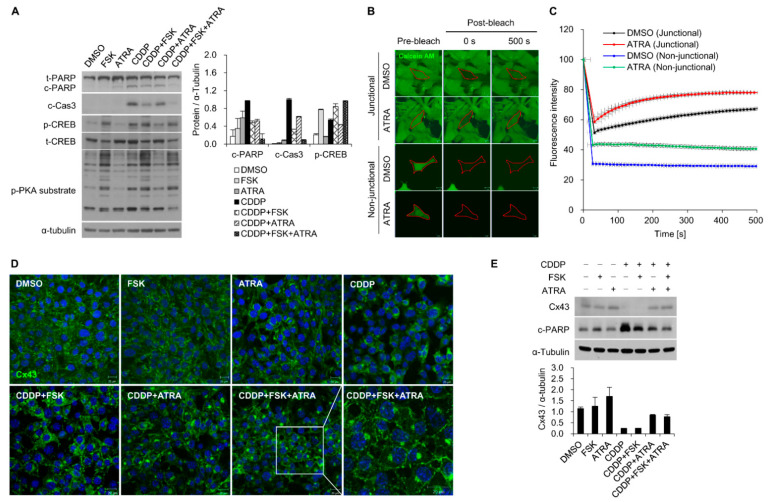
The gap junction enhancer ATRA potentiated the protective effect of FSK against CDDP-induced ototoxicity. (**A**) Western blot and densitometric analyses of PARP, caspase-3, p-CREB, and p-PKA substrate. p-CREB was normalized to total CREB, while cleaved PARP and caspase-3 were normalized to α-tubulin levels. Organ of Corti primary cells were treated with 25 µM FSK and 50 µM ATRA with or without 25 µM CDDP for 24 h. (**B**) FRAP assay of gap junction activity. HEI-OC1 cell monolayers were incubated in calcein AM (2 μM, 10 min, green) and photobleached with a laser. Recovery of fluorescence in the photobleached cells was monitored every 10 s for 500 s. The region of bleaching is indicated with red dotted lines. Representative confocal micrographs obtained prior to bleaching, immediately after bleaching (0 s), and after recovery (500 s) of HEI-OC1 cells in contact (junctional) and non-contact (non-junctional) conditions that were treated with ATRA (50 μM, 4 h). Scale bar = 10 μm. (**C**) Summary of the time course of FRAP data for at least three individual cells. For all bleached cells, the fluorescence signals present in each cell immediately before photobleaching were normalized to 100%. (**D**) Representative confocal images and (**E**) Western blot of Cx43 in HEI-OC1 cells after treatment with DMSO, FSK, and ATRA with or without CDDP for 24 h. The white square on CDDP + FSK + ATRA images indicates the magnified area. Significant differences were found between the four groups CDDP, CDDP + FSK, CDDP + ATRA, and CDDP + FSK + ATRA (* *p* < 0.05).

**Figure 3 ijms-22-06327-f003:**
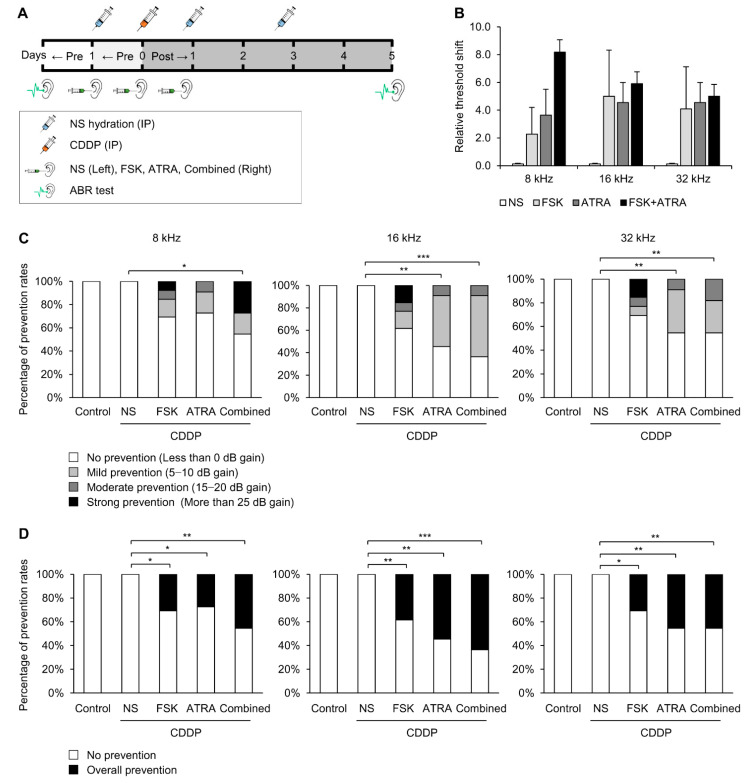
Comparison of hearing gain after FSK, ATRA, and combined treatment in terms of hearing loss prevention rates. All bar graphs show mean ± standard error (n = 11 per group). * *p* < 0.05, ** *p* < 0.01, *** *p* < 0.001. NS, normal saline; CDDP, cisplatin; FSK, forskolin; ATRA, all-trans retinoic acid; ABR, auditory brainstem response; IP, intraperitoneal; IT, intratympanic. (**A**) Experimental diagram of the in vivo animal studies conducted to investigate the effects of FSK, ATRA, and their combination on CDDP-induced ototoxicity. (**B**) Relative threshold shifts at 8, 16, and 32 kHz measured on day 5 after CDDP injection. (**C**) Comparison of hearing loss prevention rates. Hearing loss prevention rates were categorized as mild, moderate, or strong. (**D**) The overall hearing loss prevention rate is given as the sum of mild, moderate and strong prevention.

**Figure 4 ijms-22-06327-f004:**
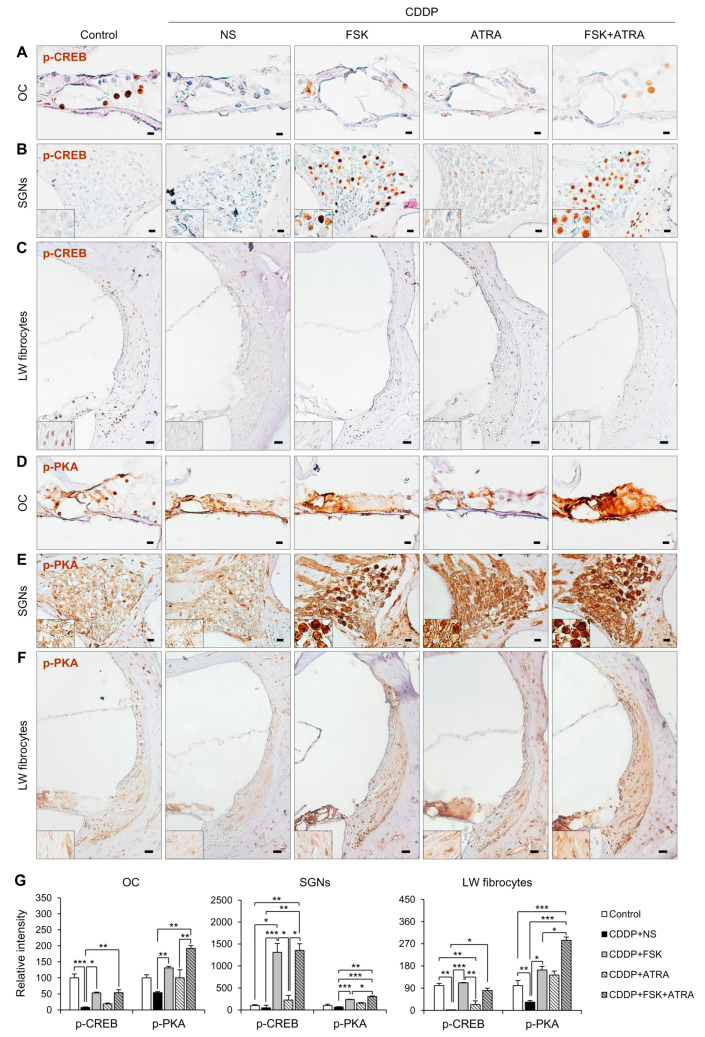
Immunohistochemical staining of p-CREB and p-PKA in rats treated with NS, FSK, ATRA, or FSK + ATRA under CDDP exposure. Representative images of p-CREB and p-PKA (brown, DAB) in the OC (**A**,**D**), SGNs (**B**,**E**), and LW fibrocytes (**C**,**F**) obtained from the middle turn of the cochlea (n = 5 per group). The cells were counterstained with hematoxylin (blue). The inset shows a magnified image of the square box. (**G**) Quantification of chromogenic intensity in the OC, SGNs, and LW fibrocytes in the middle turn of the cochlea in each group. Scale bar = 20 μm * *p*  < 0.05, ** *p* < 0.01, *** *p* < 0.001. NS, normal saline; CDDP, cisplatin; FSK, forskolin; ATRA, all-trans retinoic acid; OC, organ of Corti; SGNs, spiral ganglion neurons; LW, lateral wall.

**Figure 5 ijms-22-06327-f005:**
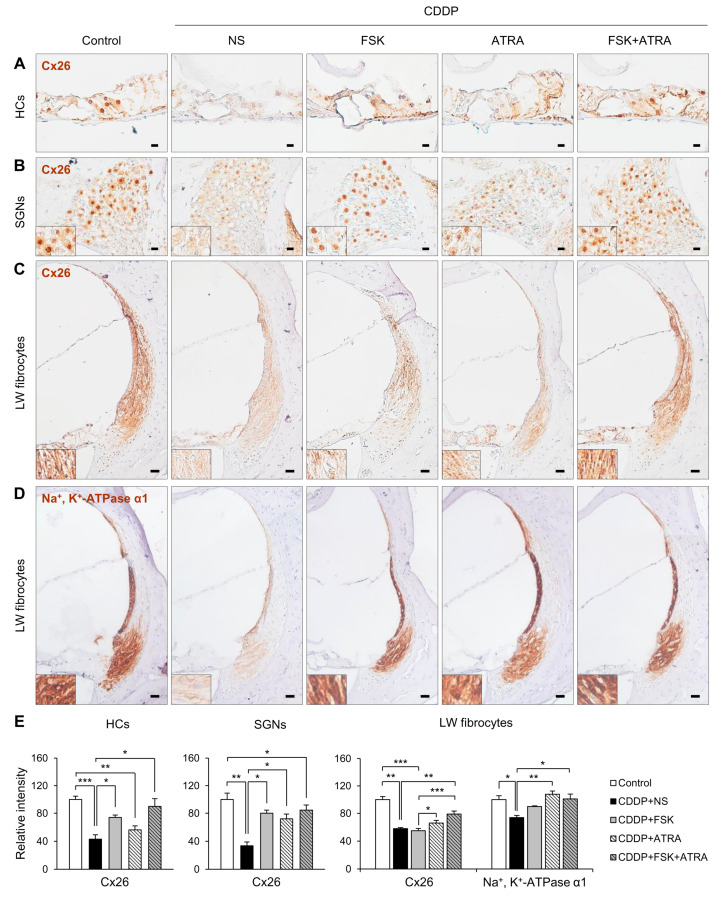
Immunohistochemical staining of Cx26 and Na^+^/K^+^-ATPase in rats treated with NS, FSK, ATRA, or FSK + ATRA under CDDP exposure. Representative images of Cx26 (brown, DAB) in the OC (**A**), SGNs (**B**), and LW fibrocytes (**C**) obtained from the middle turn of the cochlea (n = 5 per group). (**D**) Representative image of Na^+^/K^+^-ATPase in LW fibrocytes. The cells were counterstained with hematoxylin (blue). The inset shows a magnified image of the square box. (**E**) Quantification of the chromogenic intensity in the OC, SGNs, and LW fibrocytes in the middle turn of the cochlea in each group. Scale bar = 20 μm * *p* < 0.05, ** *p* < 0.01, *** *p* < 0.001. NS, normal saline; CDDP, cisplatin; FSK, forskolin; ATRA, all-trans retinoic acid; OC, organ of Corti; SGNs, spiral ganglion neurons; LW, lateral wall; Cx26, connexin26.

**Figure 6 ijms-22-06327-f006:**
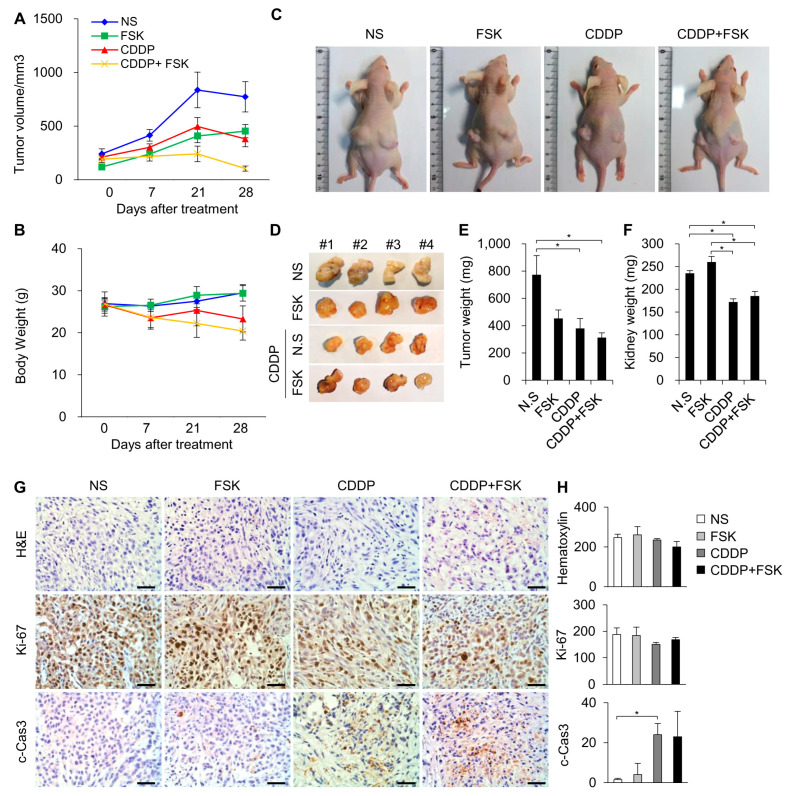
FSK does not inhibit the anti-tumor effects of CDDP in nude mice with lung cancer cell xenografts. Tumor volume (**A**) and body weight (**B**) curves for each group during drug treatment (n = 6 mice per group). (**C**) Representative mice. (**D**) Dissected tumor. Tumor (**E**) and kidney (**F**) weights after sacrifice. The bar graphs show the average tumor and kidney weights ± standard deviation in each group. (**G**) Representative images of tumors from the mice treated with NS and FSK with or without CDDP. Tumor sections were stained with hematoxylin and eosin, Ki-67, and cleaved caspase-3. The brown stained cells represent the positive cells for Ki-67 and cleaved caspase-3, which are counterstained with hematoxylin. (**H**) Quantitative analysis of hematoxylin, Ki-67, and cleaved caspase-3-positive cells in six random fields of view. The bar graph shows mean ± standard deviation. Scale bar = 40 μm. Differences of * *p* < 0.05 were considered statistically significant.

## Data Availability

The data presented in this study are available in either this article or in the Appendix A.

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
