# Peer review of "Gap Junction-Mediated Intercellular Communication of cAMP Prevents CDDP-Induced Ototoxicity via cAMP/PKA/CREB Pathway"

_ijms, 2021, doi:10.3390/ijms22126327_

Round 1
Reviewer 1 Report
After reading the manuscript my major concerns are as follows
- Page 3: The text starting on page 3 from the line 114 - up to the line 148 is duplicated. Please, delete this duplication. Please, compare the text on page 2 from the line 79 up to the page 3 line 113.
- Please, provide information about the range of doses of the tested compounds, as illustrated on Figure 1A. The doses of the tested drugs were selected based on a pilot study or from the literature.
- Legend to Figure 1B: Description of the diagram is completely different from the illustration of the results. Please, change the legend, so as to the results illustrated on Figure 1B were consistent with their description.
- Page 5 line 178: Synergistic effect between ATRA and FSK. How this synergistic effect was measured and statistically confirmed? Which statistical test was used to confirm synergy between drugs. Generally, to confirm synergy researchers must statistically prove that the observed effect considerably differs from additivity or zero interaction (for more information see: Berenbaum MC. What is synergy? Pharmacol Rev. 1989 Jun;41(2):93-141. PMID: 2692037; Tallarida RJ. Quantitative methods for assessing drug synergism. Genes Cancer. 2011 Nov;2(11):1003-8. doi: 10.1177/1947601912440575; Chou TC. Theoretical basis, experimental design, and computerized simulation of synergism and antagonism in drug combination studies. Pharmacol Rev. 2006 Sep;58(3):621-81. doi: 10.1124/pr.58.3.10.). Without any statistical confirmation researchers must not claim on synergistic interaction! Synergy must be statistically evidenced. To confirm synergy between drugs the authors should conduct experiments by means isobolography. Otherwise, the authors can ascertain that the studied drugs potentiated only their own effects. The same situation – line 193. This needs confirmation.
- Page 6 lines 215 and 216: It is not clear, which animal’s ear was injected with saline and which with FSK and CDDP. Description of the experiments suggest that only right ears were injected with both saline and FSK and CDDP. Please, correct the description. What was injected into left ear?
- Page 11, line 325 – strong synergistic effect. Please see my 4th How this strong synergy was confirmed?
- Page 12 – Discussion: line 347: The authors cannot use the term synergy without any statistically confirmed evidence. Please, replace the term synergy with potentiation.
- Syntax and grammar should be verified. Some typos should be corrected: line 241 – prevenetion; line 410 – gemcitabiene;
Author Response
Point 1: Page 3: The text starting on page 3 from the line 114 - up to the line 148 is duplicated. Please, delete this duplication. Please, compare the text on page 2 from the line 79 up to the page 3 line 113.
Response 1: We apologize for our mistake. We have delete text starting on page 3 from the line 114- up to the line 148.
Point 2: Please, provide information about the range of doses of the tested compounds, as illustrated on Figure 1A. The doses of the tested drugs were selected based on a pilot study or from the literature.
Response 2: According to the reviewer’s suggestion, we added the information of the use of doses of the tested compounds in the results section, as follows:
Page 2, line 84: “The drug doses were selected based on a pilot study and the literature.”
Point 3: Legend to Figure 1B: Description of the diagram is completely different from the illustration of the results. Please, change the legend, so as to the results illustrated on Figure 1B were consistent with their description.
Response 3: Description of the Fig. 1B was placed in a different position. We changed the legend, as follows:
Page 4, line 153: (B) (delete), --> Page 4, line 155: (B)
Point 4: Page 5 line 178: Synergistic effect between ATRA and FSK. How this synergistic effect was measured and statistically confirmed? Which statistical test was used to confirm synergy between drugs. Generally, to confirm synergy researchers must statistically prove that the observed effect considerably differs from additivity or zero interaction (for more information see: Berenbaum MC. What is synergy? Pharmacol Rev. 1989 Jun;41(2):93-141. PMID: 2692037; Tallarida RJ. Quantitative methods for assessing drug synergism. Genes Cancer. 2011 Nov;2(11):1003-8. doi: 10.1177/1947601912440575; Chou TC. Theoretical basis, experimental design, and computerized simulation of synergism and antagonism in drug combination studies. Pharmacol Rev. 2006 Sep;58(3):621-81. doi: 10.1124/pr.58.3.10.). Without any statistical confirmation researchers must not claim on synergistic interaction! Synergy must be statistically evidenced. To confirm synergy between drugs the authors should conduct experiments by means isobolography. Otherwise, the authors can ascertain that the studied drugs potentiated only their own effects. The same situation – line 193. This needs confirmation.
Response 4: We agree with the reviewer’s comment. We decided to change the “synergy” to mean “potentiation”, and have changed all expression in the manuscript, as follows:
Page 1, line 23: synergistic --> potentiating
Page 1, line 25: synergistic --> was more effective for preventing
Page 5, line 178: FSK had a synergistic effect on both increasing p-CREB and decreasing cleaved PARP --> FSK both increased p-CREB and decreased cleaved PARP
Page 5, line 192: synergistically enhanced through --> potentiated by
Page 5, line 194: ATRA, which synergistically enhanced --> ATRA potentiated
Page 6, line 208: synergistically --> synergistically (delete)
Page 11, line 321: a strong synergistic effect --> a potentiating effect
Page 11, line 324: synergistically --> enhanced
Page 11, line 327: synergistically protects auditory cells from CDDP-induced ototoxicity --> potentiates the protective effect of FSK against CDDP-induced ototoxicity.
Page 12, line 343: a synergistic --> an enhanced
Point 5: Page 6 lines 215 and 216: It is not clear, which animal’s ear was injected with saline and which with FSK and CDDP. Description of the experiments suggest that only right ears were injected with both saline and FSK and CDDP. Please, correct the description. What was injected into left ear?
Response 5: We revised the one word as follow:
Page 6, line 217: right --> left
Point 6: Page 11, line 325 – strong synergistic effect. Please see my 4th How this strong synergy was confirmed?
Response 6: We revised the “strong synergistic effect” --> “potentiating effect”
Point 7: Page 12 – Discussion: line 347: The authors cannot use the term synergy without any statistically confirmed evidence. Please, replace the term synergy with potentiation.
Response 7: The term synergy was revised to potentiation following the reviewer’s comment, as below.
Page 12, line 347: synergistically enhanced cAMP-dependent --> potentiated cAMP-dependent
Point 8: Syntax and grammar should be verified. Some typos should be corrected: line 241 – prevenetion; line 410 – gemcitabiene;
Response 8: Overall grammar and typos were verified and corrected. Thank you for your comment.
Page 7, line 239: prevenetion --> prevention
Page 14, line 406: gemcitabiene --> gemcitabine
The modified file is attached.
Reviewer 2 Report
The paper entitled "Gap junction-mediated intercellular communication of cAMP prevents CDDP-induced ototoxicity via 1 cAMP/PKA/CREB pathway" is an ambitious and important contribution to the field of ototoxicity and I appreciate the opportunity to review it.
This manuscript was well written and very thorough, both in explanation of the work and in the experimental design. The authors covered a wide range of experiments in order to better parse out the possible involvement of gap junctions in CDDP-induced ototoxicity, and their inclusion of a range of models was impressive. However, due to the ambitious nature of the paper, the results section was, at times, difficult to follow. This manuscript would benefit from reorganization of the results, as well as clearer explanation of the figures.
Inclusion of the control group in the relative ABR threshold shift figure would be helpful, even if there is very little relative threshold shift. I also recommend using a different term for "well prevention". Possibly something like "strong prevention".
A minor critique is that the font/text varies throughout the manuscript and this needs to be addressed. Additionally, there are repeating paragraphs in section 2.1 of the results. Major proofreading needs to occur before revision/acceptance.
Author Response
Point 1: The paper entitled "Gap junction-mediated intercellular communication of cAMP prevents CDDP-induced ototoxicity via 1 cAMP/PKA/CREB pathway" is an ambitious and important contribution to the field of ototoxicity and I appreciate the opportunity to review it.
This manuscript was well written and very thorough, both in explanation of the work and in the experimental design. The authors covered a wide range of experiments in order to better parse out the possible involvement of gap junctions in CDDP-induced ototoxicity, and their inclusion of a range of models was impressive. However, due to the ambitious nature of the paper, the results section was, at times, difficult to follow. This manuscript would benefit from reorganization of the results, as well as clearer explanation of the figures.
Inclusion of the control group in the relative ABR threshold shift figure would be helpful, even if there is very little relative threshold shift. I also recommend using a different term for "well prevention". Possibly something like "strong prevention".
Response 1: According to the reviewer’s suggestion, the relative ABR threshold shift figure (Fig. 3B) were revised by adding control (NS) group. We also changed “well prevention” to “strong prevention”, as below:
Page 6, line 222: well --> strong
Page 6, line 228: well --> strong
Page 7, line 238: well --> strong
Page 7, line 239: well --> strong
Page 17, line 590: well --> strong
A minor critique is that the font/text varies throughout the manuscript and this needs to be addressed. Additionally, there are repeating paragraphs in section 2.1 of the results. Major proofreading needs to occur before revision/acceptance.
Response 2: The entire manuscript was proofread according to the reviewer’s suggestion. Thank you for your comments.
*The modified file is attached.
Round 2
Reviewer 1 Report
I have only one minor concern as follows:
The authors have inserted a sentence on page 2 line 84:
Page 2, line 84: “The drug doses were selected based on a pilot study and the literature.”
After this sentence, please, cite any paper (article), from which drug doses were selected. Please, insert one reference from the literature.
Author Response
Point 1: I have only one minor concern as follows:
The authors have inserted a sentence on page 2 line 84:
Page 2, line 84: “The drug doses were selected based on a pilot study and the literature.”
After this sentence, please, cite any paper (article), from which drug doses were selected. Please, insert one reference from the literature.
Response 2: According to the reviewer’s suggestion, we added the references related to drug dose selection and revised references, as follows:
Page 2, line 84: “The drug doses were selected based on a pilot study and the literature.” à“The drug doses were selected based on a pilot study and the literature [13-15].”
Page 18, line 632:
- Mughal, W,; Martens, M,; Field, J,; Chapman, D,; Huang, J,; Rattan, S,; Dixon, I,; Huang, J,; Parmacek, M. Myocardin regulates mitochondrial calcium homeostasis and prevents permeability transition. Cell Death Differ 2018, 25, 1732-1748.
- Liu, Y,; Zhong, X,; Ding, Y,; Ren, L,; Bai, T,; Liu, M,; Liu, Z,; Guo, Y,; Guo, Q,; Zhang, Y,; Yang, J,; Zhang Y. Inhibition of voltage-dependent potassium channels mediates cAMP-potentiated insulin secretion in rat pancreatic beta cells. Islets 2017, 9, 11-18.
- Bhargava, P,; Janda, J,; Schnellmann, RG. Elucidation of cGMP-dependent induction of mitochondrial biogenesis through PKG and p38 MAPK in the kidney. Am J Physiol Renal Physiol 2020, 318, F322-F328.